# Host-Parasitoid Relationship between *Apis mellifera* (Linnaeus, 1758) and *Senotainia tricuspis* (Meigen, 1838) (Diptera, Sarcophagidae): Fly Aggression Behavior and Infestation Rates of Senotainiosis

**DOI:** 10.3390/insects14050415

**Published:** 2023-04-26

**Authors:** Gianluca Bedini, Chiara Benedetta Boni, Francesca Coppola, Simona Sagona, Matteo Giusti, Mauro Pinzauti, Antonio Felicioli

**Affiliations:** 1Department of Veterinary Sciences, University of Pisa, Viale delle Piagge 2, 56124 Pisa, Italy; gianlucabedini@virgilio.it (G.B.); chiara.boni@avanzi.unipi.it (C.B.B.); francesca.coppola@vet.unipi.it (F.C.); simona.sagona@unipi.it (S.S.); giusti.matteo@hotmail.it (M.G.); 2Interdepartmental Centre of Agro-Environmental Research “Enrico Avanzi”, University of Pisa, Via Vecchia di Marina 6, 56122 Pisa, Italy; 3Department of Pharmacy, University of Pisa, Via Bonanno 6, 56126 Pisa, Italy; 4Italian Beekeeping Federation (FAI), Corso Vittorio Emanuele II 101, 00186 Rome, Italy; 5Interdepartmental Research Centre “Nutraceuticals and Food for Health”, University of Pisa, Via del Borghetto 80, 56124 Pisa, Italy

**Keywords:** dipteran endoparasitoid, parasitization, honey bee disease, behavioral observations, infestation rate, pupation, emerging adults

## Abstract

**Simple Summary:**

*Senotainia tricuspis* is a dipterian endoparasitoid of the honey bee. It is responsible for the severe damage (called senotainiosis) of apiaries in several European, North African and Middle Eastern countries. Despite the availability of data on the infestation percentages and the increasingly growing awareness of the senotainiosis damage in beekeeping, the aggression and parasitization behavior of *S. tricuspis* towards *A. mellifera* remains poorly investigated. In this study, a description of parasitisation behavior, as well as data on the pupation and emergence of *S. tricuspis* in an apiary in the province of Pisa (Central Italy), is provided. The categories of aggression, beecatcher, chase and parasitization in the attack behavior of *S. tricuspis* toward western honey bees were identified and described. Moreover, the daily temporal pattern of the number of aggressions showed two main peaks: one during the morning hours and one in the afternoon. Data on the sinking depth of larvae and successful pupation allowed us to hypothesize that mulch and/or minimum soil tillage could prevent severe senotainiosis in apiaries.

**Abstract:**

*Senotainia tricuspis* (Meigen, 1838) is a Sarcophagid dipteran endoparasitoid of *Apis mellifera* L., and myiasis, caused by this fly, is reported in several European, North African and Middle Eastern countries. Nevertheless, very little knowledge concerning the aggression and parasitisation behavior of *S. tricuspis* toward *A. mellifera* is available in the scientific literature, and the temporal pattern of aggression remains unclear. The aim of this investigation was to describe the aggressive behavior of *S. tricuspis* and to provide data on pupation and adult emergence in order to identify further tools for the control of senotainiosis in beekeeping. Data were collected in an apiary in Pisa province (Tuscany, Italy), where observations of aggressive behavior were conducted indirectly by using a VHS camera and also directly by an observer. Four behavioral categories of the attack were described. A total of 55 aggressions, 21 beecatchers, 104 chases and 6 parasitization events were recorded with the camera. Slow-motion recording analyses of the parasitization episodes resulted in contact of at least 1/6 s between the parasitoid and the host. Through four days of direct observations, a total of 1633 aggression events were recorded. The daily temporal pattern of the number of aggressions showed two main peaks: one during the morning hours (i.e., from 10:00 to 11:00) and one in the afternoon (i.e., from 15:00 to 17:00). The morphometric data on the first-instars of *S. tricuspis* allowed us to hypothesize a penetration in the bee through its prothoracic spiracle as a modality of entrance in the host body. Third-instars successfully pupate when sinking in topsoil or clay soil, and adults emerge when left to a 4 °C overwintering period of six months. Furthermore, the high mortality rate of those larvae that did not sink and did not pupate successfully suggests that reaching a certain depth in the soil is a determining factor for larvae survival and that mulch and/or minimum soil tillage could prevent severe senotainiosis in apiaries.

## 1. Introduction

*Senotainia tricuspis* (Meigen, 1838) is a Sarcophagid dipteran occurring in Europe, North Africa and the Middle East [1,2]. *S. tricuspis* is similar to a domestic fly, with a length of 5–8 mm and is gray-black in color with a white strip between the reddish compound eyes and three black cusps on the abdomen [1,3]. *S. tricuspis* is an endoparasitoid of *Apis mellifera* L. and is responsible for senotainiosis, a syndrome that affects the bee’s flight ability that is associated with the spread position of the wings (K-wings) and which could lead to the collapse of bee colonies when the infestation rate exceeds 70% [4,5,6,7]. Lower infestations can be highly debilitating for western honey bee colonies if associated with other diseases, such as varroosis [7]. Damage to apiaries has been previously reported in Albania [8], Algeria, Jordan [2,9,10,11], Egypt [2,10,12], France [13,14], Italy [15,16,17,18,19,20], Oman [2,12,21,22], Portugal [23,24], Tunisia [14,25], Romania [26], Syria [2,25,27], Spain [28,29], Ukraine and Russia [30,31].

In central Italy, adults of *S. tricuspis* emerge during late springtime and begin to infest honey bee hives from the second half of May/early June [18]. Females are larviparous and young larvae are carried inside the uterus until deposition on the thorax of the host honey bee [3,18,32]. Once the first-instars are laid by the adult, they penetrate the honey bee thorax [16] and begin to migrate into the trachea, where they develop into second instars [19]. During this phase, second instars severely damage the host tracheal system, feeding on the bee hemolymph and muscles and leading to the host’s death [7,19]. Third instars complete their development by eating parts of the dead honey bee tissues for 4–5 days, leaving the host body and undergoing a metamorphosis in the ground, where they develop into pupae [19,32,33]. Adults start to emerge in spring after overwintering in the soil, and the larvae that reach the pupal stage during June–July develop into adults within 15–20 days, allowing a second generation in the same year [7].

Although senotainiosis is reported in several European, North African and Middle Eastern countries, the aggression and the parasitization behavior of *S. tricuspis* have been poorly investigated. Flies are reported to attack honey bees that forage on flowers [30], fly out of the hives [13,16] or fly back to the hives [18]. Furthermore, the temporal pattern of the aggression behavior remains unclear, although many authors suggest that the flies attack mainly during the hottest and brightest daylight hours [16,18,34,35,36]. Since information on the aggression behavior, pupation and biological cycle of *S. tricuspis* could provide useful elements for the control of senotainiosis in beekeeping practice, the aim of this study was to investigate the host–parasitoid relationship between honey bees and the Sarcophagid dipteran fly.

## 2. Materials and Methods

### 2.1. Biological Samples and Infestation Rates

Biological data of *S. tricuspis* were collected once every week, from June to November in 2004 and 2005, from a hive of an apiary in the Pisa province (Tuscany, Italy, 43.6515° Lat., 10.3054° Long., 0.96 m a.s.l.). The hive was arranged in sites protected from the wind currents by the wall structure of a six m high building. During data collection, the monthly mean temperatures ranged between a maximum of 24 °C in August and a minimum of 11 °C in November in 2004, and between a maximum of 24 °C in July and August and a minimum of 11 °C in November in 2005. The monthly cumulative rainfalls ranged from a minimum of 16 mm in July and August to a maximum of 129 mm in November 2004 and from a minimum of 9.9 mm in July to a maximum of 103 mm in November 2005. Following the methodology of Santini et al. [18], during each day of sampling (25 days per year), 20 honey bees were collected and conserved for 24 h in 180 mL jars. The dead bees were moved into Petri dishes for the observation and sampling of emerging *S. tricuspis* larvae. For each dead honey bee, the number of larvae inside the host body was counted. The infestation rate for each day of sampling was calculated by:$$\frac{total\;number\;of\;emerging\;larvae\;\mathrm{per}\;day}{total\;number\;of\;bees\;in\;the\;sample}\times 100$$

In August 2004, ten adult females of *S. tricuspis* were collected and weighed in order to compare the parasitoid and the host body weights. After weighing, the females were dissected under a stereomicroscope; the uterus was removed, and larvae were counted and measured.

### 2.2. Behavioral Study

An observation of the sinking behavior of the *S. tricuspis* larvae was performed by sampling 60 larvae emerging from bee bodies and placed into 6 thin transparent boxes (6 × 14 × 14 cm) containing three types of soil: sand, clay and topsoil. For each cage, 10 larvae were placed on the soil surface, and the larvae sinking depth was measured after 10 days. The pupariae that developed during the sinking phase (*n* = 25) were removed from the soil and divided into two groups. The first group, comprising 13 pupariae, was left at 4 °C for 6 months and at 25 °C for a further 3 months. The second group, comprising 12 pupariae, was left at 25 °C for 9 months. The number of emerging adults from both groups was recorded.

Observations of the behavior of adult *S. tricuspis* were performed in August 2004, directly by an observer operator and indirectly using a VHS camera Sony Hi 8, set to operate with a 1/10,000 s shutter speed. The observations were conducted from 08:00 to 19:00 for four days (19–22 August; mean temperatures range: 22–27 °C; rain falls: 0 mm; mean wind speed range: 6.9–23.7 km/h). Each set of observations was performed for 20 min every hour, following the behavior sampling method associated with continuous recording [37]. In accordance with the parsimony principle and with their intrinsic unitary consistency from the onset to the end [38], four behavioral categories were defined [7]. During the direct observations, the aggression events of *S. tricuspis* toward bees were differentiated between those toward the forager bees leaving (flying-out) and those toward forager bees entering (flying-in) the hive.

On the 27 August, two additional adult females of *S. tricuspis* were collected, marked on the thorax with red nail polish and weighed. From 17:00 to 18:00, the number of individual aggressive behaviors of the two marked flies toward the bees was recorded.

## 3. Results

### 3.1. Biological Samples and Infestation Rates

A total of 1000 honey bees were collected, of which 500 were collected in 2004 and 500 were collected in 2005. A total of 120 and 121 parasitized honey bees in 2004 and 2005, respectively, were recorded. Each honey bee hosted only one *S. tricuspis* larva. For both years, the temporal pattern of the infestation rate of *S. tricuspis* in honey bees is reported in Figure 1. In 2004, a peak of 75% of infestation on the 1st of October was recorded, while in 2005, the peak of infestation (45%) occurred on the 3rd of September.

The average weight of the *S. tricuspis* adult females was 8699 mg (SD = 1,06; *n* = 10), which represents 4.8% of the average weight of a worker bee (i.e., 180 mg [39]). 

Uteruses removed from the dissected adult females of *S. tricuspis* contained an average of 557 first-instars, ranging from 512 to 602 larvae, with a length of 800–1000 µm and a diameter of 100–130 µm.

### 3.2. Behavioral Study

The pupation depths of 60 larvae, sampled from the parasitized honey bees and placed in boxes with three different soils, are reported in Table 1. A total of 25 larvae sank successfully and developed into pupariae, with an average depth of 4.25 cm and 3.06 cm for the two clays boxes (rate of mortality = 50%) and 3.45 cm and 3.31 cm for the two topsoil boxes (rate of mortality = 35%). Pupariae placed in the sandboxes failed to sink, and only two larvae developed into pupariae (rate of mortality = 90%).

All pupariae that were left at 4 °C for 6 months and then at 25 °C for 3 further months (*n* = 13) developed into adults, while only one adult emerged from the pupariae left at 25 °C for 9 months. All the emerging adults were female. 

A total of 18 h of direct observation and 12 h of video recording on the aggressive behavior of *S. tricuspis* were performed. Through video-recording analysis, the aggressive behavior of *S. tricuspis* toward honey bees resulted and comprised four behavioral categories (Figure 2): (1)*Aggression*: the act of flying toward a flying honey bee from an ambush position.(2)*Beecatcher*: the act of flying toward a flying honey bee from an ambush position and immediately returning to the same ambush position. The term for this category was taken from the name “flycatcher”, the small migratory bird, *Muscicapa striata* (Pallas, 1764), that presents the same peculiar behavior during its predatory activity toward its prey (generally flies).(3)*Chase*: the act of pursuing a flying honey bee.(4)*Parasitization*: the act of chasing a flying honey bee, followed by contact with the host for a recorded time of four frames (1/6 s).

A total of 6 parasitizations, 55 aggressions, 104 chases and 21 beecatcher events were recorded with the camera. The slow-motion recording analysis of the six parasitization episodes revealed a time of 1/6 s (= 4 frames) of permanence of the fly on the tergal part of the body of the honey bee. During the aggression and parasitization episodes, bees apparently failed to change their flying trajectory and behavior. When the contact occurred, the fly and the bee could fly together over 50–100 cm. The analysis revealed that the fly requires a straight, unobstructed flight path to reach and parasitize the bee. 

The direct observations failed to detect the parasitization events due to the too-fast movements of *S. tricuspis,* and only aggressions were detected. During the four days of direct observations, a total of 1633 aggression events, with a peak in the morning (10:00–11:00; mean T = 28 °C) and a peak in the afternoon (15:00–17:00; mean T = 28 °C), were recorded (Figure 3). 

The number of aggression events toward the flying-out bees and the flying-in bees, recorded during the four days of direct observations, is reported in Figure 4. For each day, the results of the aggression event peaks recorded in the morning were directed toward the flying-out bees, while the peaks in the afternoon were directed toward the flying-in bees. 

The weights of the two marked flies were 15.03 mg and 15.99 mg, and the observation of aggressive behavior from 17:00 to 18:00 revealed 71 and 65 aggressions, respectively (which corresponded to 23 and 22 aggressions, respectively, in 20 min).

## 4. Discussion

The results obtained in this investigation confirmed the presence of *S. tricuspis* in Tuscany (Italy), as already reported by [15,16], with a peak infestation in early autumn for both years investigated. The differences in the rate of infestation of these two peaks (75% in 2004 and 45% in 2005) could be the result of an intrinsic variability of the population density of *S. tricuspis*, which may be altered by different climatic and environmental factors, which still remain poorly investigated [2,6,25,30,40]. The rates of *S. tricuspis* infestation in apiaries have been investigated in several European, North African and Middle Eastern countries [2,6,19,21,25,26,28,30,41,42], resulting in a high variability of infestation rates. Haddad et al. [25] hypothesized that such variations in the infestation levels between the northern Mediterranean countries and southern countries could be due to the preference of *Senotainia* for wet areas rather than dry areas and to a larger susceptibility of *Apis mellifera ligustica* (Spinola, 1806) to *S. tricuspis* attacks [25]. The differences among countries were also recorded for the flying and infestation peak periods of *S. tricuspis*, varying with the altitudes and latitudes of the investigated area [6,25,30]. Pinzauti et al. [43] found high values of *S. tricuspis* infestation in uncultivated areas rather than in cultivated areas in Tuscany, most probably due to the tillage, which kills larvae that pupate in the soil [43] or, alternatively, due to the presence of natural enemies that may reduce the fly population, as reported by Marchiori et al. [44,45,46], for other members of the Sarcophagidae family.

Each parasitized honey bee that was sampled in this investigation hosted only one *S. tricuspis* larva. This result could suggest a capacity of discrimination by the fly for already parasitized bees, i.e., each honey bee is attacked and parasitized only once by a fly. Alternatively, cannibalism may occur among larvae inside the host bees, determining the survival of only one larva. The cannibalistic behavior of larvae has already been observed in Callyphoridae dipteran *Chrysomya albiceps* (Wiedemann, 1819) [47,48], and further studies to also confirm the presence of cannibalism in *S. tricuspis* are desirable. 

This study aimed to fill the void of lack of information on the aggressive and parasitization behavior of *S. tricuspis* towards *A. mellifera*. The attack modalities of *S. tricuspis* can be divided into three different sequences, followed by a fourth behavioral event that represents the actual parasitization of western honey bees. The duration of a parasitization event (1/6 of a second, corresponding to four video frames) is very close to the limits of the human eye to perceive moving objects, and the description of such an event is possible, but only if it is recorded by a video camera and then analyzed in slow-motion. During parasitization, flies and bees fly together over a distance of 50–100 cm, and in all six video-recorded events, such contact failed to change the flight direction of the parasitized honey bee. The preservation of the honey bee’s flight movement, associated with the average weight of the fly, corresponding to 4.8% of the average body weight of honey bees (i.e., 180 mg [39]), may support the hypothesis that *S. tricuspis* hosts are unable to perceive the fly’s weight on the thorax, thus determining a lack of defense mechanism implemented by the bees. However, it was not possible from the video-recording analysis to detect if honey bee targets were already parasitized before their contact with flies and how many larvae were laid during the parasitization events.

The results of the number of aggression events during the four days of direct observation clearly indicate the presence of two main peaks of aggression, i.e., from 10:00 to 11:00 and from 15:00 to 16:00. The highest number of aggressions recorded in the morning were all toward honey bees leaving the hives, while during the same hours of observation, the number of aggressions toward honey bees entering the hives did not exceed nine events. Conversely, the peak of aggression in the afternoon was toward honey bees entering the hives for all four days of observations. These temporal patterns could be due to the trend of honey bees leaving their hives in the morning hours, determining a higher density of the same going-out directions, which is more easily detectable by flies. The same phenomenon occurs during the afternoon when honey bees tend to return to the hives, creating higher densities of the same coming-in directions. Authors have reported that *S. tricuspis* attack honey bees that forage on flowers [30], fly out of the hives [13,16] or fly back to the hives [18]. The results of the aggression behavior reported in this study could explain the differences in the temporal pattern of aggression behavior reported in the literature [16,18,36,49]. The decrease in the aggression events recorded during the middle period of the daylight hours (from 12:00 to 14:00, mean T = 30 °C) for all four days of the observations could be due to high temperatures, as well as to lower honey bee flight activity. However, in this investigation, since our observations were performed in front of the hives, and both the direct and indirect observations focused on the immediate vicinity of the hives, the decrease in the aggressions from 12:00 to 14:00 could also be due to a possible shift in the ambush location, not visible to the observer performed, by *S. tricuspis*.

Red-marked flies, observed through direct observations, performed a lower number of attacks towards western honey bees compared to the unmarked flies (22.5 aggressions on average vs., e.g., on the 20 August, 84 aggressions). Such a lower number could represent the real number of individual aggressions that a single fly can perform or, alternatively, could be due to the weight of the marking paint, which had a negative influence on flies’ flight activities.

To the best of our knowledge, no data concerning the number of *S. tricuspis* larvae contained inside the female uteruses and their dimensions are available in the literature. The results reported in this investigation represent a preliminary dataset on the number of larvae contained in the uterus, which ranged from 512 to 602 for each adult female of the *S. tricuspis* analyzed (*n* = 10). The first-instars were measured, and the results obtained could suggest a new hypothesis on the penetration modalities inside the honey bee’s body. Since the larvae presented a diameter of 100–130 µm, which is compatible with the dimensions of the prothoracic spiracles (185 µm wide, 731 µm long [50]) occurring on the bee’s thorax, a simple transit of larvae through these openings could be hypothesized. Santini and Pinzauti [18] provided an alternative penetration pattern, suggesting that larvae, through their own motion, create lacerations in the keratinous tissues between the head and the thorax and enter the honey bee’s body [18]. In this context, further investigations on the penetration modalities of *S. tricuspis* are desirable.

In this investigation, the pupation depth of *S. tricuspis* third-instars in different types of soil was measured. Although Piazza and Marinelli [42] reported a preference for *S. tricuspis* larvae to pupate in sandy soil, the results reported in this investigation suggest an unsuitability of the sand (larvae mortality rate = 90%) for the sinking mechanism performed by larvae. These results also suggest that sinking is a determining factor for the pupation and survival of larvae of *S. tricuspis*. 

When exposed to low temperatures for six months, 100% of the pupariae (*n* = 13) developed into adult females, while at 25 °C conditions, only one puparium developed into an adult successfully. These results, combined with the high mortality rate of larvae in sand soil, suggest that mulch and/or the minimum soil tillage associated with the use of sandy soil in the immediate vicinity of the apiary could be used to contribute to senotainiosis control. A similar conclusion was deducted by Pinzauti et al. [43], who reported higher infestation rates of senotainiosis in uncultivated areas. Nevertheless, further studies on the sex ratio and reproduction, as well as studies focusing on the minimum duration of the successful overwintering of *S. tricuspis*, allowed us to detect new methodologies for the control of senotainiosis in apiaries, which are desirable.

*S. tricuspis* can lead to the collapse of a honey bee family when it reaches a rate of infestation above 70% [6]. In cases of severe senotainiosis, the use of chromotropic sticky traps could reduce the levels of infestation in apiaries, where white traps placed on the roof of the hive can successfully attract adults of *S. tricuspis* [51,52].

## 5. Conclusions

The infestation rates of *S. tricuspis* show a variability of the temporal pattern, depending on the altitude and the latitude. In the Tuscany region at low altitudes (0–500 m a.s.l.), the peak of infestation occurs in early autumn, and during the morning hours, flies seem to attack the honey bees when leaving the hives, while in the afternoon, they tend to attack the honey bees returning to the hives. The aggressive behavior of *S. tricuspis* is modulated in four different behavioral events, and contact with the honey bees occurs only during the parasitization event, which lasts 1/6 s. Even if slow-motion video-recorded analyses of the event were performed, it was not possible to detect if the targeted honey bees were already parasitized before the contact with flies occurred and how many larvae were laid during the parasitization events. Nevertheless, measurements of the first-instars allowed us to hypothesize a penetration in the honey bee body through its thoracic spiracles. Third-instars successfully pupate if sinking occurs in topsoil or clay soil, and adults emerge when a 4 °C diapause of 6 months occurs, despite the minimum duration of exposure to the same cold period remains unknown. Moreover, the high value of the mortality rate of larvae in sandy soil allows us to suggest that mulch and/or minimum soil tillage could prevent severe senotainiosis in apiaries.

## Figures and Tables

**Figure 1 insects-14-00415-f001:**
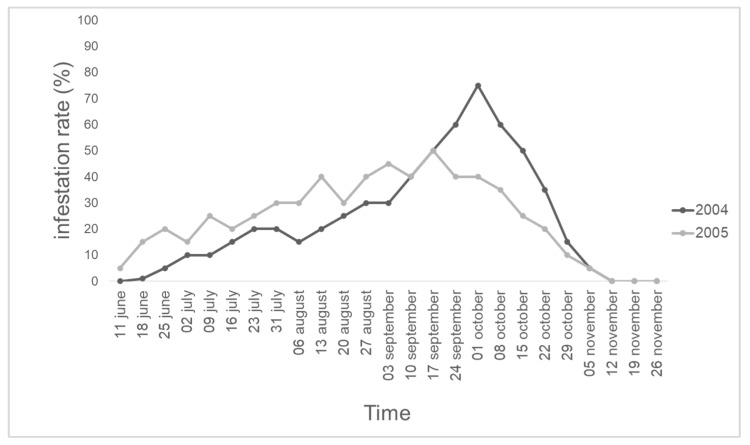
Temporal pattern of *S. tricuspis* infestation rates in honey bee hives, from June to November 2004 and 2005, in an apiary in Pisa province, Italy.

**Figure 2 insects-14-00415-f002:**
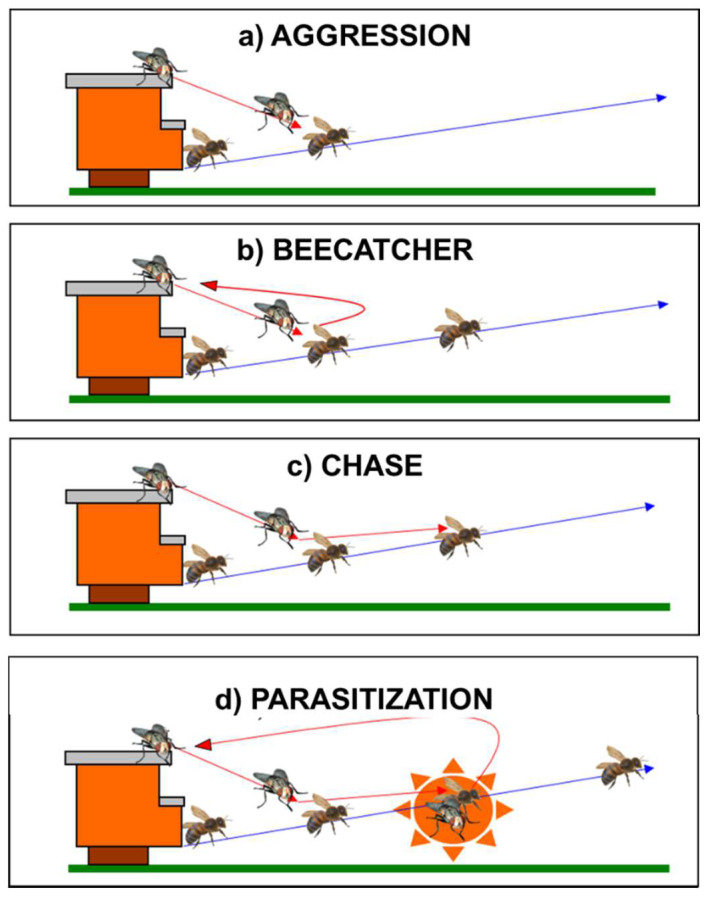
A sequence of a parasitization event, detected through video-recording analysis, showing the four different behavioral categories: (**a**) Aggression: the act of flying toward a flying honey bee from an ambush position; (**b**) Beecatcher: the act of flying toward a flying honey bee from an ambush position and immediately returning to the same ambush position; (**c**) Chase: the act of chasing a flying honey bee; (**d**) Parasitization: the act of chasing a flying honeybee followed by contact. Red arrows indicate the flight directions of *S. tricuspis*. Blue arrows indicate the flight directions of *A. mellifera*. (Drawings from [7]).

**Figure 3 insects-14-00415-f003:**
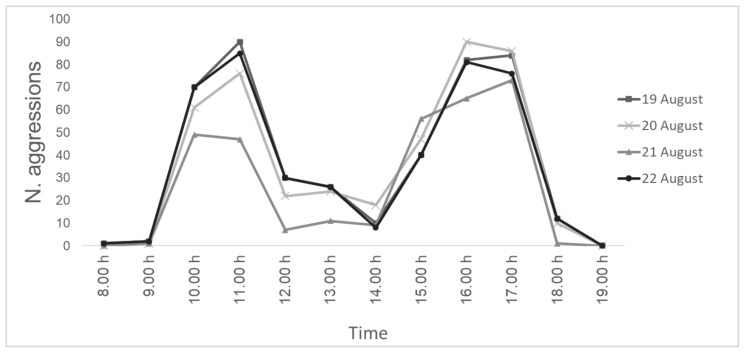
Number of aggressions of *S. tricuspis* toward honey bees detected during the four days (19–22 August) of direct observations, from 8:00 to 19:00, in Pisa province, Italy.

**Figure 4 insects-14-00415-f004:**
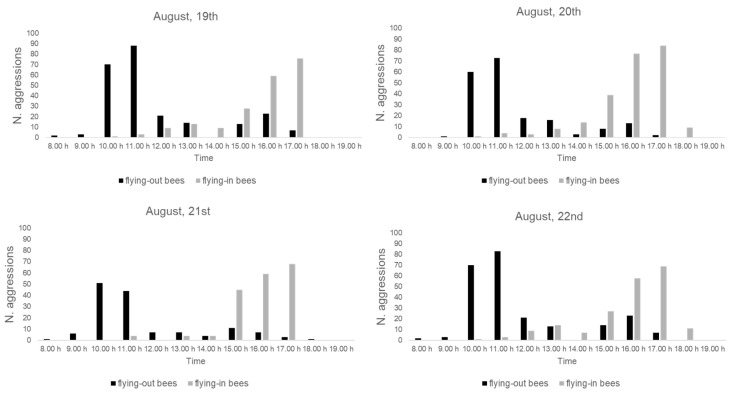
Number of aggressions of *S. tricuspis* toward flying-out honey bees (black histograms) and flying-in honey bees (gray histograms) during the four days (19–22 August) of direct observations, from 8:00 to 19:00, in Pisa province, Italy.

**Table 1 insects-14-00415-t001:** Pupation depth (cm) of *S. tricuspis* larvae in six boxes containing clay (*n* = 2), topsoil (*n* = 2) and sand (*n* = 2) and the number of developed pupariae for each box.

Type of Soil	Boxes	No. Developed Pupariae (Tot = 25)	Pupation Depth (cm)							Average Pupation Depth (cm)
Clay	1	4	3.0	3.4	4.7	5.9	-	-	-	4.25
	2	6	1.1	2.2	2.3	3.7	4.0	5.1	-	3.06
Topsoil	1	6	1.5	2.0	2.8	3.9	4.8	5.7	-	3.45
	2	7	0.8	1.2	1.2	3.9	4.0	5.6	6.5	3.31
Sand	1	0	-	-	-	-	-	-	-	-
	2	2	0	0	-	-	-	-	-	0

## Data Availability

The data presented in this study are available on request from the corresponding author.

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
