# Peer review of "Host-Parasitoid Relationship between Apis mellifera (Linnaeus, 1758) and Senotainia tricuspis (Meigen, 1838) (Diptera, Sarcophagidae): Fly Aggression Behavior and Infestation Rates of Senotainiosis"

_insects, 2023, doi:10.3390/insects14050415_

Round 1

Reviewer 1 Report

The manuscript presents original data on Senotaina tricuspis elucidating the aggression behavior of the dipterand endoparasitorid and provide data on pupation and emerging. Even if the data refer to observations made in 2004 and 2005, they are still of considerable interest and fill a knowledge gap. 

Reviewer 2 Report

Manuscript ID: insects-2313474 entitled” Host-parasitoid relationship between Apis mellifera (Linnaeus) and Senotainia tricuspis (Meigen) (Diptera, Sarcophagidae): fly aggression behaviour, infestation rates and control of senotainiosis” by Gianluca Bedini et al. submitted to section: Insect Societies and Sociality is an important valuable study on this fly endoparasitoid of honey bees.

Biological data of S. tricuspis were collected in an apiary in the Pisa province, Italy. Through analysis of video-recordings and field observations authors identified and descripted four different behavioral interactions between the fly and the foraging honey bees. Authors also made observations on fly pupation depth in relation to soil type. All in all, the information is novel adding to our knowledge on this endoparasitoid and its effect on honey bees in that region. I added my notes to attached PDF and recommend consideration for publication with major revision. My concern on lacking meteorological data during the survey and video observation are important and should be included in this article. Also, the article does not support a claim of a control method as implied in the title.

Round 2

Reviewer 2 Report

Revised manuscript: insects-2313474 has improved. Authors revised according to suggested changes by reviewers which warrant consideration for publication in the present revised form.